# Application of Deep Reinforcement Learning in Traffic Signal Control: An Overview and Impact of Open Traffic Data

**Martin Gregurić [1,*,†,‡], Miroslav Vujić [1,‡], Charalampos Alexopoulos [2] and Mladen Miletić [1]**

[1]  Faculty of Transport and Traffic Sciences, Department of Intelligent Transportation Systems, University of Zagreb, 10000 Zagreb, Croatia; mvujic@fpz.unizg.hr (M.V.); mmiletic@fpz.unizg.hr (M.M.)
[2]  Department of Information and Communication Systems Engineering, University of the Aegean, 83200 Karlovasi, Greece; alexop@aegean.gr
*  Correspondence: mgreguric@fpz.unizg.hr
†  Vukelićeva 4., Zagreb HR10000, Croatia.
‡  These authors contributed equally to this work.

**Abstract:** Persistent congestions which are varying in strength and duration in the dense traffic networks are the most prominent obstacle towards sustainable mobility. Those types of congestions cannot be adequately resolved by the traditional Adaptive Traffic Signal Control (ATSC). The introduction of Reinforcement Learning (RL) in ATSC as tackled those types of congestions by using on-line learning, which is based on the trial and error approach. Furthermore, RL is prone to the dimensionality curse related to the state–action space size based on which a non-linear quality function is derived. The Deep Reinforcement Learning (DRL) framework uses Deep Neural Networks (DNN) to digest raw traffic data to approximate the quality function of RL. This paper provides a comprehensive analysis of the most recent DRL approaches used for the ATSC algorithm design. Special emphasis is set to overview of the traffic state representation and multi-agent DRL frameworks applied for the large traffic networks. Best practices are provided for choosing the adequate DRL model, hyper-parameters tuning, and model architecture design. Finally, this paper provides a discussion about the importance of the open traffic data concept for the extensive application of DRL in the real world ATSC.

**Keywords:** deep reinforcement learning; adaptive traffic signal control; multi-agent systems; intelligent mobility; deep neural networks; open traffic data; big data

## 1. Introduction

In the last decade, significant progress has been made in the development of the advanced traffic control methods due to persistent problems with intense congestions and their negative impact on sustainable mobility. The most prominent breakthrough in traffic control has been made by introducing methodologies that do not require an exact empirical model for complex traffic flows interactions. Those methodologies have enabled a self-adaptation ability, pro-active strategies, and coordinated control in the traffic control that was not possible with control approaches that heavily dependent upon the accuracy of the traffic models. Most of those methodologies are based on Machine Learning (ML) so its effectiveness depends on the computational power and extensive data that accurately and comprehensively describe the current and past traffic states.

Currently available traffic datasets can be categorized as *big data* since they are even at this point very large, complex, and originate from numerous different sources that can generate large quantities of data in short time intervals. These characteristics make those datasets very difficult or impossible to

process by the traditional ML approaches applied in traffic control without the need for an extensive pre-processing in the form of adequate data structuring and feature extraction. The aforementioned has led to the need for a novel approach that will retain the ability of classic ML in performing prediction or classification based on the non-linear function approximation, but simultaneously it must remove the need for manual feature extraction.

The deep learning is the latest ML approach that is based on the learning representations of data. This approach attempts to model high-level abstractions in the data by using multiple processing layers with complex structures or otherwise composed of multiple non-linear transformations [1]. The model which contains multiple processing layers with complex structures based on the concept of the human cerebral cortex is known as Deep Neural Network (DNN). The DNN integrates feature extraction, and classification (or prediction) process in a single framework by using information-dense input datasets. Those raw datasets are usually formatted as tensor-like labeled data entries. The digestion and learning process of such large datasets by using complex DNN is computational expensive so it requires extensive GPU processing power (or use of clusters and cloud computing) in order to increase the learning stability and induce sufficiently fast convergence towards the desired goal. In the recent decade with the advancements in GPU processing power, deep learning has become the most prominent research direction in a ML-driven traffic control problems, mainly due to its scalability potential and automatic feature extraction directly from raw traffic data.

This paper is focused on traffic signal control since the impact of congestions is most noticeable in the dense urban traffic networks. The most used machine learning methodology for traffic signal control is Reinforcement Learning (RL). This approach is used due to its relatively simple algorithmic structure which is based on the real-time "*trial and error*" experiments where the errors are used to compute quality estimations of each trial. Those quality estimations are computed and stored in the table-like data structures in accordance with the trials defined as state–action pairs. That approach enables support for on-line learning which is a necessary feature for the self-adaptation to the constant temporal and spatial fluctuation in traffic demand, and at the same time, it does not require large historic datasets. The previously mentioned table-like structure represents the data containers based on which quality function can be computed. With the large possible state–action space such as in the case of traffic control systems, those table-like structures can suffer from high-dimensionality. The DNN models are used as the efficient quality function approximators. That approach eliminate a need for a mentioned table-like structure in the RL framework. The integration of RL and DNN is known as the term Deep Reinforcement Learning (DRL).

An overview of DRL frameworks for traffic signal control provided in this paper represents an extension of traditional RL approaches for the same purpose described in [2]. Furthermore, this paper provides an analysis of the several most representative DRL frameworks used for traffic signal control. Those frameworks are analyzed with respect to their learning hyper-parameters, DNN model design, and optimization algorithms. All those parameters are evaluated according to characteristics of the use case scenarios used to validate each of the analyzed DRL control frameworks. Additionally, special emphasis is set on the multi-agent DRL frameworks which enable coordinated control over several intersections in the traffic network. The main focus is set on detailed analysis regarding the raw intersection data pre-processing which must be in line with the DNN model digestion requirements. This analysis is necessary for giving the design guidelines for high-level image-like data formatting in the context of Open Traffic Data [3]. The exchange of mentioned standardized data-structures in the Open Traffic Data framework is especially important for establishing interoperability and flexible scalability between the various DRL approaches applied for the holistic traffic signal control.

The remainder of this paper is organized as follows. Section 2 looks into the background of the RL with a description of its drawbacks and theoretical approach for its augmentation with the use of deep learning. This section is concluded with an overview of the latest DRL frameworks commonly used in the traffic signal control. Section 3 explains the benefits of using DRL based algorithms over the traditional approaches for traffic signal control. Special emphasis is set on the DRL based

algorithms that can conduct coordinated control over several intersections in the traffic networks. Section 4 provides deeper insight into the current approaches for the design of the traffic signal control which is based on the DRL. The proposed framework based on Open Traffic Data for raw traffic data pre-processing and sharing in the context of deep learning and large intersection networks is described in Section 5. Finally, this paper is concluded with the discussion and conclusions.

## 2. Reinforcement Learning

The RL can be described as the discrete, stochastic control process in which future states depend on the current states and taken actions. Transitions between states are governed by the policy function $\pi(s_t)$ in a given time step as defined by the Markov Decision Process (MDP). The RL is the most used traditional ML approach for traffic signal control since it does not need an exact model of the stochastic traffic flows behavior in a network of intersections. The RL based agent is able to gain a knowledge troughs the learning process and iteratively model the dynamics of the controlled traffic environment just by interacting with it [4].

Actions are usually generated randomly at the beginning of the learning process; thus, initially, there is no knowledge about the control policy. Therefore, there is a need for extensive exploration of state–action space. The likelihood of choosing random actions decreases over the learning time, thus the action is increasingly being selected according to the learned policy function as the learning process progresses through time. In other words, the probability that a policy function reproduces a good solution increases during the learning process, thus it is logical to exploit it more frequently. The problem of modeling the relationship between the random exploration of state–action space and exploitation of learned policy function during the learning process is known under the term "*exploration vs. exploitation*" trade-off. The $\epsilon - greedy$ algorithm is the most used approach for modeling the "*Exploration vs. exploitation*" trade-off problem in learning process for choosing actions [5]. The RL algorithms should always retain the ability to conduct exploration in order to achieve control robustness to potential new states in a controlled environment. This feature of RL algorithms is especially important for control over the traffic flows which are prone to sudden spatiotemporal fluctuations.

The traditional RL agent receives a scalar reward after performing an action in the controlled environment. The goal of RL agents is to learn an optimal control policy; thus, the discounted cumulative reward is maximized via repeated interaction through the learning process with its environment [4]. Supervised and unsupervised machine learning approaches are usually one-shot, myopic, considering instant reward, while RL is sequential, far-sighted, considering long-term accumulative reward [6]. One of the most used value-based or off-policy RL approaches for traffic control is known as the Q-learning algorithm [7]. It is commonly used due to its efficiency and simple logical architecture which does not require an explicit definition of policy function. Its long-term state–action value function is parameterized and updated using the step-wise experience [8]. In other words, it iteratively updates the optimal Q (Quality)-value by using the newly received learning sample $(s_t, a_t, s_{t+1}, r_t)$ according to the *Bellman* equation defined as

$$Q^*(s_t, a_t) = Q(s_t, a_t) + \alpha_n(r_t + \gamma \max_{a' \in A} Q(s_{t+1}, a') - Q(s_t, a_t)), \tag{1}$$

where $Q^*(s_t, a_t)$ is a new Q-value, $Q(s_t, a_t)$ is current Q-value of state–action pair, $\alpha_n$ is learning rate, $r_t$ is the reward received from the environment, $\gamma$ is discount rate which can be static or change over time in order to model process in which the earlier rewards are worthier than the rewards in the future, $\max_{a' \in A} Q(s_{t+1}, a')$ is the maximum expected future reward, and $A$ represents possible action space. After every learning iteration, Q-values are updated and then stored in a Q-table (or *look-up* table) in accordance with the corresponding state–actions pairs. At this point, it is possible to conclude that Q-values for each state–action pair represent their expected future rewards derived as the long-term accumulative rewards.



Since the Q-learning is the off-policy based RL algorithm it is characterized by efficient updating according to the *bootstrapped* sampling of Experience Replay (ER). However, its update is based on a one-step temporal difference, so the good convergence relies on a stationary MDP transition, which is found to be less likely in Adaptive Traffic Light Signal Control (ATSC), as remarked in [9]. As a contrast, in policy-based RL methods, such as REINFORCE (also known as *Monte Carlo* variant of policy gradients), the policy is directly parameterized and updated with sampled episode return so it does not use a value function [10]. The REINFORCE is an on-policy RL algorithm so it can make the non-stationary transition within each learning iteration. This is useful for stochastic and continuous action space, but it is hard to define a scoring system which will adequately evaluate a learned policy since the rewards are computed after the end of the episode. For example, one episode represents one traffic simulation, and the one-step denotes one control action during this simulation. The on-policy algorithm requires many episodes and a well-defined scoring function to converge towards the desired results. This is the reason this algorithm is not extensively applied in the traffic control systems. An Actor–Critic logical architecture integrates a value-based approach as its Critic part and policy-based approach as its Actor part in a single RL framework usually by using two separate models for each of its parts. This framework is popular since it provides additional reduction of bias and variance during the policy-based model learning by using another model for parameterization of the value function which provides an adequate assessment of the learned policy function [9].

### 2.1. Drawbacks of Reinforcement Learning

The traditional Q-learning algorithm in every learning iteration requires a *brute force* grid search trough the entire Q-table. Exponential enlargement of the Q-table in traffic control problems is induced due to the discretization of the action space since the traffic control decisions must be made in equal time intervals. The discretization of traffic control actions is also important for easier implementation in simulation environments. Numerous macroscopic traffic parameters (e.g., speed, density, incoming vehicles to each intersection bound, queue length, and their values in several previous time steps) can be arranged in the vector-based state representation while the actions are mainly modeled by the finite number of scalars (e.g., a predefined number of traffic signal phases denoted by indexes). It is possible to conclude that Q-table in the majority of the ATSC problems for the large urban networks can be affected by high-dimensionality. Visitation of each state–action pair in table-like structures by using the classical grid-search approach can become computationally infeasible and this problem is known as *curse of dimensionality* [11]. To tackle problems with high-dimensionality of continuous space, numerous non-linear function approximation methods are considered. The idea of function approximation is to avoid computing the exact Q-value by calculating its approximation which covers the whole state–action space. Usually, this is done by fine partitioning of state–action space. This approach computes scalable fitting over continuous states and design heuristic state features. Function approximation methods are also suitable to create an estimation of the Q-values in regions of the state–action space that were not visited during the learning process [12]. Application of function approximation in RL produces significant results only in cases where the states are low-dimensional and handcrafted with linear value or policy functions. In the context of machine learning, the following approximators are used: decision trees, Artificial Neural Networks (ANN), and k-Nearest-Neighbor (kNN) regression methods. Application of the traditional ANN as the function approximator has shown problems in the stability of RL [13].

### 2.2. General Deep Reinforcement Learning

Recent publications [11,13] propose an integration of deep learning methodologies with RL under the term DRL. The main purpose of this integration is the policy function approximation by the use of DNN based models. In early studies [14], DNNs models for DRL are represented by a deep Stacked-AutoEncoders (DeepSAE), while the recent models are based on Convolution layers (Conv), which are stacked along with the flatten one and several Fully Connected (FC) layers. The flatten



layer converts two-dimensional features computed by convolution layers into the one-dimensional vector suitable as the input for FC layers. Those DNN models are known as the Convolutional Neural Networks (CNN) since they contain convolution layers that conduct feature extraction from the image-like inputs by simulating biological convolution processes in human visual cortex [15]. Additional Convolution layers in the CNN model allows the development of features from the features extracted in previous layers by the process of sub-sampling, transforming low-level features of the data to high-level ones. This process can potentially increase the overall CNN performance [16]. The FC layers in CNN have the role to provide a structural condition classification and eventually assess the probability for the execution of each possible action according to the passed inputs to the CNN. The configuration of the CNN model used in DRL does not contain pooling layers which are mandatory for example in image classification. Those layers are not needed in CNN setup for DRL since the exact positions of traffic fluctuation in the image-like inputs are essential for traffic control. The CNN model applied in DRL for traffic control along the all mentioned type of layers can contain Long Short-Term Memory (LSTM) layers based on Recurrent Neural Network (RNN) architecture. They have the ability to process data sequentially and keep hidden state through time. The LSTM layers are used in CNN for traffic control in [17], while the Residual Networks (ResNet) as the unique setup for CNN are used in [6]. The ResNet architecture for CNN models introduces a skip connection among a group of layers (ResNet blocks) in order to jump over some layers and reuse activations from a previous layer from which jump is made until the adjacent layer learn its weights. The addition of LSTM layers and the introduction of ResNet architecture in the CNN models is proposed in order to tackle *vanishing gradient* problem and induce a more stable learning process. Furthermore, it is possible to introduce the Batch Normalization (BN) process in the form of an added Normalization layer between layers in DNN models. The mentioned normalization has to be done separately for each dimension of the input layer in order to adjust and scale its activations. The application of BN can improve the stability and speed of any DNN model, as proposed in [18].

The integration of complex DNN models with the RL can enhance its learning capacity in order to tackle complex tasks such as control over the stochastic traffic flows at the numerous intersections [9]. In general, the following RL approaches are extended by the use of DNN models for the sake of traffic control: value-based, policy-based, and Actor–Critic methods [19]. Deep Q-learning (DQL) is an augmentation of the vale-based Q-learning algorithm where $Q^*(s_t, a_t)$ is approximated by the use of DNN model with weights $\theta$. This setup is described as:

$$Q(s_t, a_t; \theta) = Q^*(s_t, a_t) \tag{2}$$

The DNN digest input states arranged as the image-like matrix or one-dimensional vectors which may contain several channels if it is needed. If the input state contains several channels then it can be considered as the tensor—represented by the stack of several image-like matrices. Usually, each mentioned matrix is populated with different types of traffic parameters describing the same traffic scenario in the current action interval. The DNN model has a predefined number of outputs that correlate with the number of possible actions. The Q-value is computed for each action after every learning iteration. The DNN model is learned to minimize the expected squared error (also known as *loss*) between the predicted Q-value and the target Q-value, as described by the following equation:

$$L = \mathbb{E}\left[(r_{t+1} + \gamma \max_{a' \in A} Q(s_{t+1}, a') - Q(s_t, a_t; \theta))^2\right] \tag{3}$$

Target Q-values are usually computed based on the separate target DNN model, which has the same architecture as the Q-value DNN model. The weights of the target DNN model are updated periodically by copying weights $\theta'$ from the Q-value DNN model (which updates its weights during each learning iteration). Keeping the fixed weights in target DNN model for a predefined period of time ensures a temporally static Q-value target. That eliminates the moving

target problem and consequentially provides stabilization of the DQL learning process. Additionally, the described approach reduces drastic oscillations in the control policy design upon small changes of the Q-value [20]. The use of the two DNN models in the DQL framework is commonly known under the term Double DQL. It is shown that Double DQN effectively mitigates over-estimation of rewards in noisy controlled environments such as traffic flows in complex traffic networks, and therefore improves its overall performance [5]. The target Q-value is described as

$$Q_{target}(s_t, a_t) = r_t + \gamma Q(s_{t+1}, \arg \max_{a' \in A} Q(s_{t+1}, a'; \theta); \theta'). \tag{4}$$

In Double DQL framework, the output of Equation (4) replaces the $\max_{a' \in A} Q(s_{t+1}, a')$ from Equation (3). Another important part of the general DQL is Experience Replay (ER). The ER is used as the *memory buffer* for storing the experience tuples $\{s_t, a_t, r_t, s_{t+1}\}$ during the observation phase in DQL learning process. The observation phase starts with the operational work of DQL, while the end of this phase means that the ER is full. At the end of the observation phase, it is possible to sample Mini-Batches from ER. Mini-Batches are used as the learning sets of inputs for the Q-value DNN model. The ER has a predefined size; thus, when the new experience tuple is added, the last stored tuple is removed. Furthermore, many studies use *reward clipping* procedure which scales and clips the rewards in a specific range (commonly $[-1; +1]$). The described procedure is done to avoid the boosting of weights when back-propagating [20]. Functional scheme of general DQL can be seen in Figure 1.

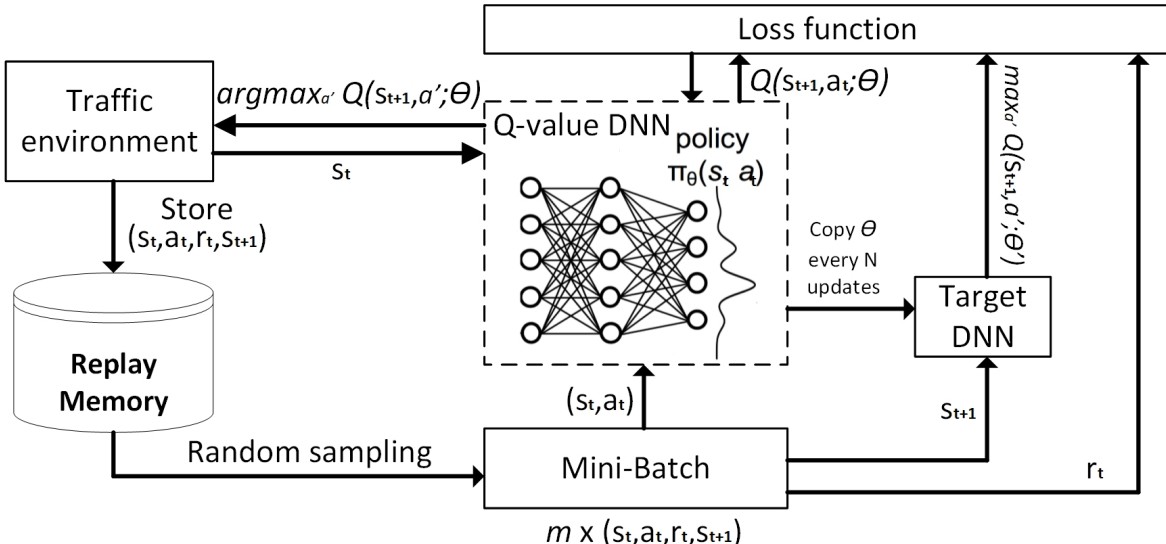

**Figure 1.** General DQL framework scheme.

The most commonly used algorithms for the update of DNN weights are based on the first-order gradient optimizers such as Stochastic Gradient Descent (SGD) algorithms [14,16], and RMSprop [21,22], but it is possible to include a more advanced optimization method such as ADAM [23] and AdaBound [24]. The weights must be optimized in every learning iteration to induce the minimization of Equations (3).

### 2.3. Advanced Approaches for Deep Reinforcement Learning

The Rainbow algorithm is a colloquial name for the DQL framework which employs a full spectrum of DQN extensions: previously described Double Q-learning, Prioritised Replay Memory (PRM), Dueling DNN, multi-step learning, distributional RL, and noisy nets. In the traffic signal control problems, the following DQL extensions from the Rainbow algorithm are commonly applied:

- Prioritized Experience Memory (PER) allows the RL based agent to consider state–action transitions with the different frequency that they are experienced [25]. It increases the replay probability of samples stored in RM that have a high Temporal-Difference (TD) error, and therefore possible high impact on learning convergence. The TD error is computed based on the difference between the current and targeted Q-values. Mentioned prioritization can lead to a loss of diversity. This problem can be alleviated by introducing the stochastic prioritization, and biased outputs can be corrected by importance sampling, as described in [26].
- Dueling Deep Q-Networks (DDQN) represents the special architecture of DNN models used in DQL. The Q-value is estimated according to the value of the current state and each action's advantage of taking this action *a* in state *s*. The value of state $V(s; \theta)$ is the overall expected reward in the case of taking probabilistic action in the future steps. Advantage, denoted by $A(s, a; \theta)$, is computed for all possible actions in accordance with the given state and with the main task to describe how important particular action is to the value function compared to the other actions. The final Q-value is computed by summing the value *V*, and advantage *A*. The dueling architecture is able to improve the performance of DQL as it can be seen in the following studies [5,27].

Recently, Actor–Critic architecture based on the discrete state representation is the most used approach for the design of the DRL frameworks. This approach can be understood as the architectural methodology which augments on-policy algorithms such as REINFORCE with the ability to make an update at each time step by using two DNN models. One of these two DNN models perform the role of Critic which approximates the value function for each learning step. This value function is adopted instead of the total session reward function used in the policy gradient. The main idea of Critic is to evaluate the impact of the current state on bringing the agent control policy closer to its long-term objective. This assessment is based on the TD error presented as:

$$TD = r_{t+1} + \gamma V(s_{t+1}) - V(s_t), 0 \leq \gamma \leq 1 \tag{5}$$

where $\gamma$ is s the discount factor that represents the difference in importance between future and instant rewards, $r_{t+1}$ is the instant reward, and $V(s_{t+1})$ is the state value which indicates how well the state *s* at time *t + 1* is based on the long-term objective [28].

The second DNN model has the role of an Actor. It is used as a policy function approximator or. in other words, it governs how agents select actions. Commonly, both DNN models are trained in parallel with the different sets of weights that must be optimized independently. A novel approach in Actor–Critic DNN model architecture includes sharing the lower layers (usually convolution layers) between the Actor and Critic parts in the same DNN model. Those two parts are differentiated at the higher layers of the same DNN model usually by the use of LSTM layers, as presented in [17].

The general idea of the Actor–Critic architecture is to deliver current state representation from the controlled environment to both models. Policy (Actor) model computes the action according to the mentioned state and consequentially affects the environment. Changes in the environment induce a new state for the next action interval. This new state is evaluated with the appropriate reward. Computed reward and the new state is passed to the value (Critic) model. Critic computes the Quality (Q-value) of transition from taken action to that particular state, while Actor updates its policy parameters (weights of its own DNN model) by using Q-value computed by Critic. The actor computes the next action according to the updated Q-value and new state, and at the same time Critic model updates its own wights. Conceptual scheme of the Actor–Critic architecture in DQL applied at the two signalized intersections can be seen in Figure 2:

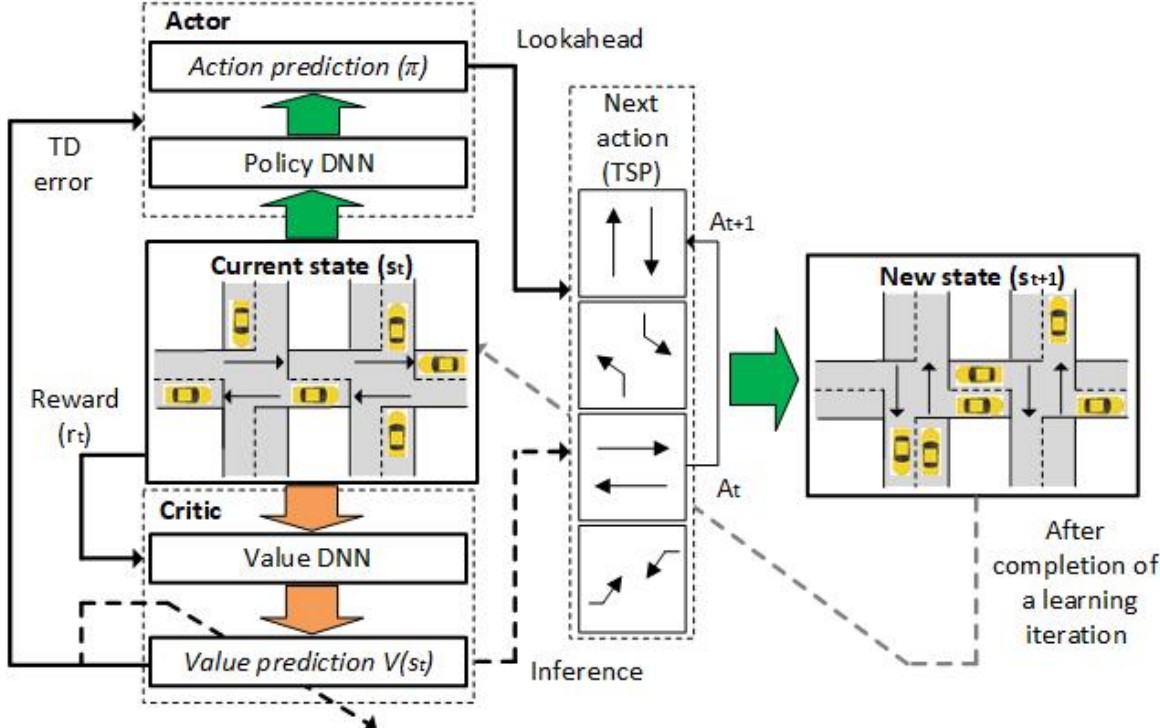

**Figure 2.** Conceptual scheme of the DQL Actor–Critic architecture applied to two signalized intersections.

There are two possible modifications of Actor–Critic architecture to reduce the variability of the value-based method and to increase its learning stability. Both are based on the advantage function application instead of the value function, and they all have a focus on parallel learning in the multi-agent framework.

- Asynchronous Advantage Actor–Critic (A3C) approach is based on the central agent with global DNN model parameters. Its workers are copies of the Actor part of this agent; thus, this approach does not use ER since it requires a lot of memory. In A3C, there is only one Critic that learns the value function while multiple copies of actors are learned in parallel as his workers. Each worker in the A3C framework is executing in parallel in different instances and each of them is periodically asynchronously synced with the global DNN model. Due to independent exchange data between workers and the global parameters, there can be an issue related to the policy inconsistency among some workers.
- Advantage Actor–Critic (A2C) approach implements the coordinator module between workers and global DNN model parameters, which waits for all workers to finish their segment of experience before updating the global DNN model. This approach enables all workers to synchronously start with the same policy. In A2C, all workers are the same and they have the same set of weights since all workers are updated at the same time. This approach requires several versions of the environment that must be executed in parallel. Additionally, one worker should be assigned for each of them. Since the traffic flows are described by complex spatiotemporal datasets there is a danger that MDP may become non-stationary if the agent only knows the current state [9]. Furthermore, it is infeasible to input all historical inputs to A2C so it is common to include LSTM layers in its DNN models, which maintains hidden states in order to memorize short history [29]. Such a setup of A2C approach makes its learning process more cohesive and faster.

However, DQL frameworks still suffers from the poor data-efficiency. That attribute limits its possible real-world application and reduces its scalability potential [30]. The multi-step learning approach in [17] is used to build an Actor–Critic framework for controlling the large network of

intersections. In the context of TD error computing, machine learning is usually thought in terms of an average value of many multi-step returns with differing lengths, and are often associated with eligibility traces. Furthermore, this approach is known as the multi-step return procedure since it can be spectated in the terms of individual *n*-step returns with their associated *n*-step backup. Each of these individual backups is refereed as an *atomic backup*, whereas the combination of several *atomic backups* of different lengths creates a *compound backup* [31].

In 2017, Uber AI Labs published several papers that are oriented towards the optimization of the DNN model by the use of evolutionary algorithms. This approach is known as Deep Neuro-Evolution. One of the previously mentioned papers [32] presents a study which introduced the concept of using the Genetic Algorithm (GA) from the domain of evolutionary inspired algorithms, in order to optimize/evolve weights of the DNN model in the context of DRL framework. The scalability of GA enables the parallelization of optimization computations over many CPU which results in much faster learning convergence of the DNN model compared to the traditional gradient-based methods. In the traffic control, this approach has been applied for the cooperative control between commonly used motorway control methods (ramp metering, differential speed limit control, and lane change control), and has initially shown better results compared to the traditional DQL. It is a policy-oriented approach with multi-agent structure and novel Knowledge Sharing Graph Convolutional Networks (KS-GCN) for the generation of coordinated actions between different motorway control methods [33]. Potentially, this approach could be a new direction in DQL design dedicated to ATSC.

## 3. Adaptive Traffic Light Signal Control Based on DRL

ATSC algorithms at the beginning of their development were based on *rule-based* methods. The *rule-based* methods were firstly implemented within the fixed signal time strategies, e.g., changing the predefined signal plans in accordance with the characteristic period of the day (non-peak and peak-hours, night- and daytime, working days and weekends, the season in the year, etc.). The heuristic *rule-based* methods such as Longest-Queue-First (LQF) or *Greedy policy* conduct traffic light phase switching in accordance to the measured traffic demand at each intersection bound [34]. Those types of algorithms provide improvements over the aforementioned algorithms since they take into account the information about the local traffic scenario by processing the data from sensors installed at the controlled intersection. It is noticed that those ATSC algorithms have reduced effectiveness in dissolving traffic congestions which are affecting urban traffic environments. Those environments are characterized by the strong fluctuations in the increased traffic demand over the day [35,36]. Those poor results are the product of the modeling process which is mostly based on common traffic scenarios without the inclusion of human errors and sudden increase in traffic demand due to various social reasons. The stochastic nature of traffic flows is tackled by the RL based algorithm such as Q-learning since they do not require a model of the controlled system. That control approach has shown constraints regarding computational efficiency, learning stability, and scalability [37–39]. The core problem with the Q-learning algorithm is linearly parameterized function approximation, which requires a set of carefully handcrafted features from sensory data to perform optimal control [34]. The use of the DNN model in the RL context enables the feature extraction directly from the raw traffic data, which allows a more comprehensive ATSC independent of the Q-table usage and human errors related to feature extraction. Even early results achieved by the application of a DRL approach in ATSC for one isolated signalized intersections are promising.

### 3.1. Traffic Signal Control on a Larger Scale

The poor scalability of DRL frameworks dedicated to ATSC originates from their RL roots. The main problem is an extremely large discrete action space which emerges in the case when one wants to gain control over the larger number of intersections. Training a centralized DRL agent for control over several signalized intersection is still infeasible for the large scale traffic signal problems, and finding a feasible solution to that problem is still an open area for research [40,41]. In the case

of large scale traffic networks, it is computationally infeasible to train one DNN model for each signalized intersection. Furthermore, it is very unlikely that the resulting policy would be globally optimal without the coordination between local learning/optimization processes [34]. The various Shallow-ANN (SANN) models presented in [42] were the first non-DRL approaches used to address the aforementioned problem. Those models were based on a lower number of hidden layers and they used only macroscopic traffic variables such as a number of vehicles at each bound of the controlled intersection as its inputs. Each SANN model was assigned to one intersection. Those models had two shortcomings. The first addresses the need for manual identification of the general traffic features and arrangement of those features within fixed time frames. The second implies that the weights are updated according to the most usual daily traffic patterns. Thus, every sudden fluctuation in the traffic demand which diverges from those patterns can reduce its efficiency.

The introduction of coordinated Multi-Agent RL (MARL) framework was the robust approach for holistic control over several intersection compared to previously used methodologies [43]. The MARL frameworks are mostly based on the Q-learning algorithm. MARL frameworks can be designed based on: (1) decentralized architecture that is based on the dynamical partitioning of the traffic networks into smaller regions where each RL agent control one of those regions; (2) transfer learning techniques; and (3) allocation of one simple RL agent for each intersection. A problem with decentralized architecture is the assumption that global network-level action-value function is simply a linear summation of regional action-value functions [34]. One of the first approaches for MARL design is described in [44] where the migration from single- to multi-intersection control was made by the use of transfer planning and the *max-plus* coordination algorithm for inter-agent optimization. The DQL agent was firstly trained on two intersections; then, by the use of transfer learning, it was extended up to the four intersections. The shortcoming of that approach was the complex reward function which is hard to implement in the real world, and the inability of efficient scaling up to the larger number of intersections.

In the third MARL approach, an agent conducts local observation and limited communication with other agents usually by distributing the global Q-function over the local agents. This is done by the use of coordination rule which governs *trade-off* between the optimization level and scalability rate, or it is possible to introduce the Independent Q-learning (IQL) approach. In the IQL approach, each local agent learns its own policy independently and enable modeling of other agents as the parts of the environment dynamics [9]. The drawback of this approach is related to the appearance of the non-stationarity environment from the point of view of each agent since the controlled environment is affected by the actions of other agents which themselves also learning [45].

That approach can be extended by the use of DQN for RL in the form of Independent DQN (IDQN). In that approach, each agent observes its nearby environment (known as the partial state), then computes individual action, and finally receives a group reward that is shared among all agents. Furthermore, the IDQN approach in [25] achieves learning over the multiple heterogeneous agents by the use of Dueling Double Deep Q-Network (3DQN) for each agent. Approaches that are based on IDQN are prone to the RM relevance reduction because the data generation dynamics in the local agent's RM no longer reflect the current dynamics in which learning is done [45]. The problems with IDQN convergence stability induced due to the mentioned drawback can be alleviated by the introduction of two approaches proposed in [9]. The first is known as *fingerprinting*, which is based on information about neighborhood policies to improve the observability of each local agent thus each local agent has more information about the regional traffic distribution and usage of cooperative strategy. That approach consequently reduces the non-stationarity nature of the multi-agent environment [25]. The second is based on the inclusion of a spatial discount factor with the main goal to weaken the state and reward signals from other agents. The most advanced IDQN technique for the deep MARL control over the several traffic intersections is the Independent A2C (IA2C) approach which is based on the Actor–Critic architecture for DQN.

The most prominent issue with the ATSC over the larger number of intersections is related to the feasible processing of input traffic data that is continuously sampled by the numerous traffic detectors. The Deep Deterministic Policy Gradient (DDPG) as the one type of off-policy DRL is proposed in [20] as the conceptual approach that can tackle the large scale ATSC problem which relies on the aggregated data from the road traffic sensors. It combines the Actor–Critic approach with the Deterministic Policy Gradient (DPG). Deterministic policy as opposed to stochastic is approximated by an ANN-based Actor $\pi(s; \theta^\pi)$ usually designed with a fully connected Multi-Layer *Perceptron* (MLP) and *Leaky* ReLU activation function. Thus, its actions depend on the state of the environment $s$ and has weights $\theta^\pi$. The size of the first layer in the Actor model is defined by the number of detectors in the traffic network and the number of available TSPs, while the final layer size corresponds to the number of actions. The DDPG off-policy nature combined with the aforementioned structure of the Actor model can provide a framework for continuous traffic data digestion from numerous sensors on a large traffic network. That structure enables the computation of actions in the form of the vector with $N$ components. Each of those components is a real number that has a scaling effect on the duration of each TSP with the main purpose to preserve the duration of the total cycle. Those scaled values for TSP duration are stored in the phase adjustment matrix. They are coded in the final layer known as "Phase adjustment". The Actor network is updated according to the applied chain rule to the loss function, and the update of weights $\theta^\pi$) is based on gradient loss. The second separate model based on fully connected layers is implemented within the Critic part which computes the value $Q(s, a; \theta^\pi)$ according to the current state. It is updated by the means of the *Bellman* equation as is the case with the DQN. The mentioned study does not provide results regarding the comparable traffic parameters but only shows favorable convergence of reward under the traffic scenario with 43 intersections.

Furthermore, Tan et al. [34] proposed a decentralized-to-centralized architecture for traffic signal control problems in a large traffic network. The main idea in this approach is to tackle large-scale control problems by transferring learned base models into the sub-regional grids. The first step is to conduct a partitioning of the large traffic grid into several sub-regional grids of the same topology. Each sub-region is paired with a single DRL agent for the local optimal policy learning within a sub-regional grid. The DRL agent is created according to the *Wolpertinger* Actor–Critic architecture which uses MLP model with ReLU activation functions instead of CNN, and Deep Deterministic Policy Gradient (DDPG) inspired approach for learning. Deep collaboration between locally computed minimal regional costs expressed trough traffic congestion is done by a hierarchical architecture based on the added global dense layer which concatenates the latent states of all local agents in order to form a global action-value function (or Q-function) for the entire controlled traffic network. The same study concludes that it makes more sense that a global RL agent in ATSC should be learned based on the global Q-value function rather than local/regional rewards aggregation. In Table 1, it is possible to see most representative approaches for ATSC design based on the DRL framework that are chronologically listed. In the same table, improvements are denoted according to the first control methodology listed in the *compared against* column.

**Table 1.** The most representative DRL frameworks for ATSC design.

| Paper | Year | DRL Method | Number of Intersections | Compared against | Control Strategy | Improvements |
|-------|------|-----------|------------------------|-----------------|-----------------|--------------|
| [17] | 2019 | Regional A3C + PER [2] | 42 | Hierarchical MARL, Rainbow DQL, Decentralized multi-agents [1] | Acyclic TSP selection and TSP duration computation | 8.78% lower Average Delay |
| [9] | 2019 | Stabilised IA2C [3] | 30 | IA2C, IQL-LR, IQL-DNN | Acyclic TSPs with fixed duration | 63.7% lower Average Delay |
| [34] | 2019 | Hierarchical regional A2C [6] | 24 | Regional DRL | Acyclic TSPs with fixed duration | 44.8% lower Waiting time |
| [46] | 2019 | 3DQN + PER | 1 | Actuated and fixed time controller | Acyclic TSPs with fixed duration | 10.1% lower Average Delay |
| [18] | 2018 | ResNet based A2C | 9 | Actuated controller | Cyclic fixed TSP switch | 16% lower Waiting time |
| [20] | 2017 | DDPG | 43 | Q-learning algorithm | Cyclic TSPs with computed duration | No data |
| [21] | 2017 | DDQN + ER | 1 | LQF, fixed time controller | Cyclic TSPs with fixed duration [4] | 47% lower Overall Delay |
| [16] | 2016 | DDQN + ER | 1 | STSCA [5] | Acyclic TSPs with intermediate TSPs | 82% lower Overall Delay |
| [14] | 2016 | DeepSAE + RL | 1 | Q-learning algorithm | Two TSPs with dynamic duration | 14% lower Overall Delay |

[1] The decentralized multi-agent approach is based on distributed constraint optimization. [2] Multi-step return and Off-policy A3C. [3] It is stabilized by fingerprinting technique and spatial discount factor. [4] The intermediate TSPs are dependent on the current phase and the chosen actions are used due to safety reasons. [5] It is compared to Shallow Traffic Signal Control Agent (STSCA) based on shallow ANN. [6] Each Agent uses *Wolpertinger* Actor–Critic MLP architecture and DDPG algorithm.

## 4. Design of the DRL Algorithm for the Traffic Signal Control

To design the DRL algorithm for traffic signal control, it is necessary to assess the use case model with respect to the traffic network complexity. After that step, one must conduct deep analysis regarding the legislative-technical constraints which must be incorporated into the algorithm design. The results of the mentioned analysis are used to select an appropriate DRL framework for ATSC. Additionally, it is necessary to select adequate architecture for the DNN model, which will be used in selected DRL framework along with the optimization algorithm selection. In Table 2, a detailed configuration of the most representative DRL frameworks used for ATSC is shown. The next step is to tune hyper-parameters by using a grid search procedure in the context of simulation experiments. The final step is to select appropriate state and action representations. That decision highly depends on the selected DRL framework, available data structure, and traffic network complexity. In that decision, one must keep in mind the state–action space size and available computational power. Finally, it is mandatory to adequately model a reward function which will steer the learning process towards the desired goal.

It is important to emphasize that prior to real-world implementation all mentioned DRL methods applied for ATSC are trained and eventually evaluated in the traffic simulators. Traffic simulator can

be constructed based on three distinctive traffic models: microscopic, mesoscopic, and macroscopic. Macroscopic models compute only aggregated macroscopic traffic parameters at a high level [47]. Since those models cannot provide raw traffic data (e.g., position of each vehicle in network) they are not suitable for DRL. Simulators based on microscopic traffic models such as Simulation of Urban Mobility (SUMO), PARAMICS [14], Aimsun, and PTV VISSIM simulate each vehicle behavior independently. These simulators can be used with DRL based algorithms since they provide raw data for each vehicle during the whole simulation run. The SUMO is the most used microscopic simulators applied for ATSC algorithms based on DRL [17,21,48,49]. Its DFROUTER tool can heuristically compute a list of routes and a list of vehicles associated with each route according to the network-wide real-world traffic data [47]. The mentioned tool enables resembling the real traffic distribution which can increase the simulation accuracy and consequently provide better learning convergence of the DRL frameworks applied for ATSC. Simulation of the complex traffic network with the microscopic model can be computationally expensive due to numerous vehicles for which behaviors must be computed independently by using CPU. The overall computational cost is even higher since it is necessary to run a demanding deep learning process in parallel with the complex traffic network simulation. The slow simulation of complex traffic networks can be partially tackled by using simulators based on mesoscopic models. Those models simulate road traffic at the level of individual vehicles, but with an aggregated behavior on links. This approach can significantly increase simulation speed but somewhat reduce its accuracy. The overall performance of mesoscopic models for multi-intersection control can be improved by introducing hybrid microscopic–mesoscopic traffic simulations.

## 4.1. State Representation

The DNN, as the essential part of DRL, reacquires a pre-processing step for obtained raw data. These data must be structured as the image-like matrices (2D matrices) or sounds-like vectors (i.e., 1D signals) to be digested by the DNN input layer (usually, it is convolution layer). Transformation of the obtained raw traffic data into the mentioned data structures can be difficult since the natural representation of the traffic flow measurements is done by the use of a labelled graph. The common approach for mentioned data transformation is to *pixelate* or segmentate the surface of the road (and sometimes its nearby road surrounding) into the small segments, partitions, or cells. The next step is to assign obtained raw traffic data to those segments with respect to their measurement location [5,13,25]. The height of each segment is usually specified by the width of the traffic lane (or several lanes which are close to each other), and its length depends on the technique that is used for obtaining the traffic data. When these data contain parameters of each vehicle such as position, acceleration, speed, etc., as described by the early applications in [4,16,50], the length of a segment corresponds to the average or minimum length of the vehicle, as illustrated in Figure 3a. Furthermore, the same road segmentation approach can be used for traffic data such as the number of passengers per road segment [13] and the direction of the moving vehicles [17], or to conduct raw vehicle speeds post-processing in the form of their normalized values. It is also possible to use a larger uniform length of a segment if the traffic data are aggregated to some extent, e.g. the number of halting vehicles, occupancy of road segments, cumulative waiting time per segment, and percentage of the vehicles that have achieved maximum allowed speed [25]. The above-mentioned road segmentation is illustrated at Figure 3b. It is possible to conclude that both approaches conduct spatial discretization of each intersection bound in a *cellular automata* fashion. Furthermore, a novel approach based on the aggregated traffic data and variable road segmentation is proposed in [51], which is illustrated in Figure 3c. Incoming lanes at each intersection bound are divided into segments with different sizes. The further the segment is from the stop line, the longer it is since it is less important for the traffic control. This approach enables longer lane length coverage for each intersection bound [51]. The segment is marked as occupied if it is at least one vehicle in it.

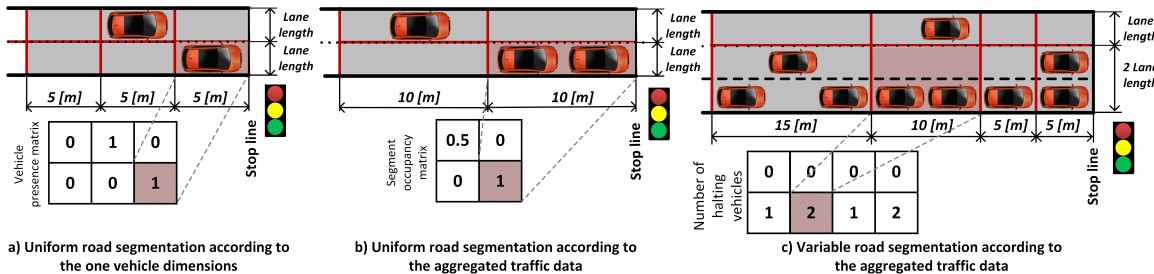

**Figure 3.** Illustration of general approaches for road segmentation in image–alike matrices.

Additionally, several image-like matrices can be stacked as the single multi-channel input sequence or tensor-like object for the DNN model digestion. It is possible to stack several image-like matrices for the current time step, but with different content regrading the traffic data or it is possible to stack several image-like matrices with the same content but measured in several consecutive time intervals within the one action step. There are several methodologies for coding traffic scenarios through the use of image-like matrices. Figure 4a illustrates the most basic approach where the segmented quadratic region corresponds to the intersection and its nearby surrounding. Segments that correspond with the intersection bounds are highlighted in grey. If they are occupied by vehicles, they have value 1; otherwise, they have value 0. The drawback of this approach is the appearance of the redundant parts in the image-like matrix denoted by the white color in the aforementioned figure. They represent areas that are not part of the particular intersection road surface thus they are not significant for learning

Shabestary and Abdulhai [13] proposed segmentation into the smaller uniform segments of each intersection bound per traffic lanes starting from the stop line at each bound. All those segmented lanes are stacked below each other into the one image-like matrix according to their intersection-bound affiliation. This image-like binary matrix contains zeros and ones, depending on whether the segment is occupied by the vehicle or not, as illustrated in Figure 4b. Both mentioned illustrations represent only one channel in a possible multi-channel image-like input sequence or *stack*. For example, Shabestary and Abdulhai [13] used the number of passengers per each segment as the second *stack* channel. The same study addresses the problem in which the image-like matrices represent traffic scenarios that might have a different meaning for each element in a single row or column, in comparison to the real image of the traffic scenario. For example, one element in a column might show a vehicle waiting on the north-bound movement and the next element in the same column might represent a vehicle waiting on the east-bound movement. This problem is tackled by the approach in each every traffic lane represented by the one row in image-like input is analyzed separately by the one-dimensional filters in the input convolution layer that are moved for one row in a vertical direction. Additionally, this approach can be used to evaluate a set of traffic lanes in one direction independently of the others by using two-dimensional filters, for which the height corresponds to the number of lanes, and filters should be moved vertically in image-like input for that same number of lanes.

The most recent approach presented in [46] is inspired by the definition of Discrete Traffic State Encoding (DTSE). It proposes a Discrete Time Traffic State Encoding (DTTSE) method which defines the state by using the event driven data. Unlike the previously described approaches, it uses traffic detector to create two different vectors. The first vector contains a binary representation of the vehicle-detection event existence in each discretized time step, while the other vector records the detectors occupancy at each time step [46]. An additional vector is used for storing the green light indication for each lane. Specifically, for each discretized time step, the ratio between green, yellow, and red light duration is stored. Those vectors are adequately transposed and stacked one below the other to form an image-like matrix, as shown in Figure 4d. Additionally, this matrix is extended by the set of mentioned joint vectors from the other traffic lanes at the same intersection. Each mentioned image-like matrix is created for one pre-defined time interval.

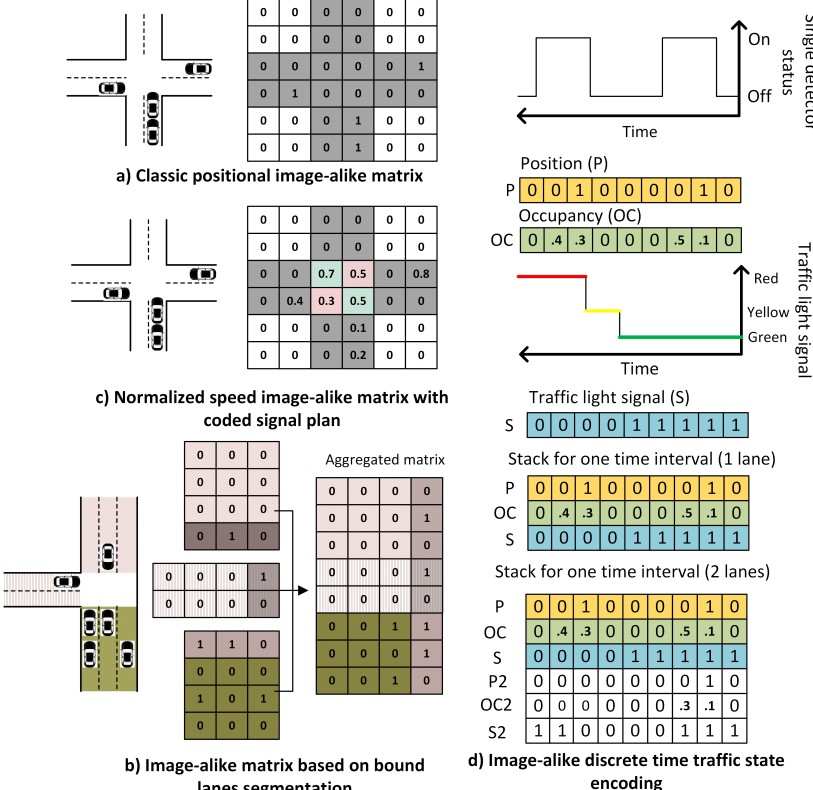

**Figure 4.** Most used traffic state configurations for one intersection in the context of deep learning.

Most of the approaches for state representation in coordinated control over several intersections require information about the current traffic lights configuration. That information can be placed at the center of the image-like binary matrix, as illustrated in Figure 4c). Ideally, the information about current traffic light configuration would be an extra layer to the state space, with binary features for each traffic light color. However, this increases the size of the state space significantly and leads to memory problems induced by large RM size, as well as slower computation during the learning process [44]. In [50], traffic data are obtained from the different sensors and formatted into a two-dimensional *HxW* tensor. More precisely, traffic data collected at the time step *t* are stored into a *triple* $\langle C, H, w \rangle$, where *C* is the number of channels which represents the traffic data such as halting vehicle number, and mean speed of vehicles, while *H* and *W* denote the height and width of single-channel, respectively, expressed as the number of segments. The sub-state of overall state tensor represents one intersection in the traffic network in which each bound is expressed with two segments. Each of those segments denotes one set of traffic lanes per each direction at particular intersection bound and they contain aggregated traffic data with respect to channel affiliation regarding traffic parameter. The complete setup of the mentioned state representation with emphasized sub-states can be seen in Figure 5a.

A novel approach presented in [9] is based on the DTSE and WAVE-like state representation for the application of a multi-agent DQL control framework in the complex traffic networks. The network of intersections is represented as the grid in which each link between nodes (intersections) denotes the traffic flow which can belong to the four possible time-variant traffic flow groups. Intersections can be connected with two or one lanes with predefined speed limits [9]. States for the time interval *t* and intersection *i* are defined by the following expression:

$$s_{t,i} = \{w_{wait,t}[l], w_{wave,t}[l]\}_{j,i \in \varepsilon, l \in L_{i,j}},\tag{6}$$

where *l* is each incoming lane at intersection *i*, $w_{wait,t}[s]$ is the measurement of the cumulative delay achieved by the first vehicle, $w_{wave,t}[veh]$ is measure for total number of approaching vehicles in

each incoming lane at distance of 50 m to the intersection according to Chu et al. [9], where each agent dedicated for intersection *i* communicates to the edge *j* neighbor $j, i \in \varepsilon$. The described state representation concept for a large traffic network is illustrated in Figure 5b.

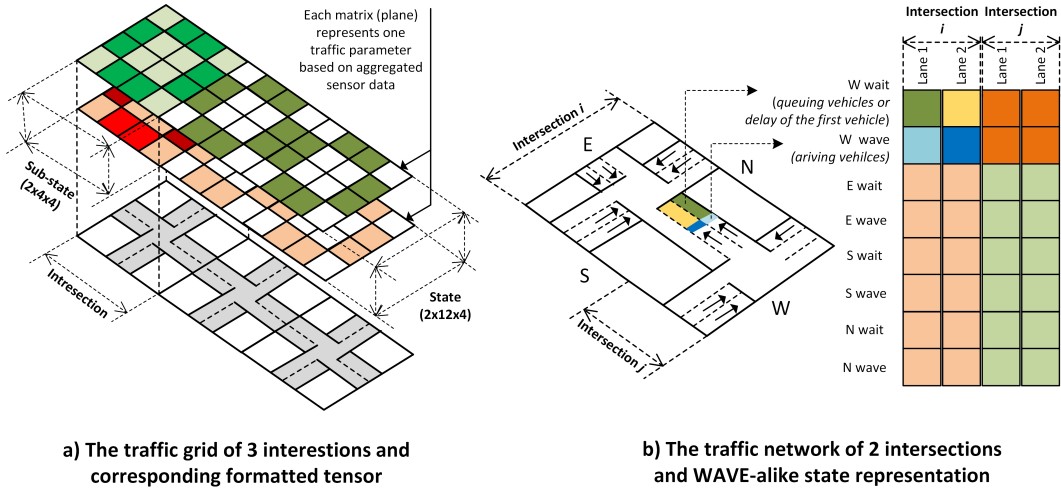

**a) The traffic grid of 3 interestions and corresponding formatted tensor**

**b) The traffic network of 2 intersections and WAVE-alike state representation**

**Figure 5.** Most used traffic state configurations for the large network of intersections.

## 4.2. Action Representation

Actions for the DRL approach applied in ATSC are configured to tackle two major problems related to traffic signal plan design: (a) scheduling strategy for Traffic Signal Phases (TSP); and (b) computation of each TSP duration. Most of the current DRL approaches applied in ATSC design use a discrete action space. That approach ensures easy implementation of the mentioned traffic signal control approach in simulation and real-world environment. The TSPs are usually denoted by the numerical indexes in the DRL framework. Compared to the active one, all other TSPs are in the red state. Scheduling of the TSPs sequencing can be fixed/cyclic (e.g., [18,21]) or acyclic (e.g., [46]). In cyclic TSP sequencing, it is possible to conclude that there are only two possible actions: (1) switch to the next phase in the predefined sequence order; or (2) stay on the same TSP. In the DRL environment that control strategy is known as the binary control decision since there are only two possible outputs of the DNN model. Furthermore, this approach usually requires additional intermediate TSP with a yellow and red light when the currently active TSP is about to change. However, the binary control decision does not clearly indicate which specific traffic streams the agent can control in the case of a multi-phase signalized intersection [52]. In binary control decisions, it is possible to get different rewards for similar state–action pairs what can lead to non-stationary distributions and consequently induce instability in the learning process.

In the acyclic TSP approach, it is necessary to index each TSP with a unique value, since the agent chooses a TSP from predefined TSP space [9,18]. In this case, TSPs scheduling is done exclusively according to the current state without any need for following predefined TSP sequencing. Furthermore, one TSP can have numerous versions of itself regarding its green light duration. Action space in the aforementioned case contains a set of predefined feasible green light duration setups for each TSP, as described in [9]. Since there is no need for computation of each TSP green light duration this approach is suitable for the DRL frameworks which evaluates each TSP modification as the one possible action. Furthermore, there is usually an additional yellow phase with a fixed time duration which is activated if the traffic light is switched from green to red and vice versa. The drawback of this approach is an extensive enlargement of the state–action space, which can dramatically increase computational cost. Furthermore, this approach in some cases can violate the traffic regulations which proscribe a fixed phasing sequence.

Duration of each TSP can be presented by *n*-tuple, where *n* denotes the number of TSPs. Each element of this tuple represents one TSP, and value of this element represents duration of green light in that particular TSP. Furthermore, these values can represent scaling factors for each existing TSP what preserves total duration of the cycle [20]. Yang et al. [17] introduced a significance of the distance between adjacent intersections. The impact of interactions between different intersections will weaken as the distance becomes larger between them. The same study proposes a concept of Adjusting Matrix of Traffic Signal Phase Control (AMTSPC). Firstly,$MTSPC_t$ matrix for all agents' control action at the time step *t* is given as follows $MTSPC_t = [A_{1t}, A_{2t}, ..., A_{Nt}]^T$ where *N* is the number of agents. Adjusted action matrix $MTSPC_{it}$ is created by applying the impact coefficient computed as $(1 - \frac{d_{ik}}{D})$ at the particular row of $MTSPC_t$, where $d_{ik}$ is the distance from intersection *i* to the adjacent intersection *k* and *D* is a radius within which the other intersections are relevant for particular intersection *i*.

It is also needed to emphasize that most of the current DRL frameworks for ATSC do not consider the problem related with breaking the normal routing rules. The traffic lights have to be synchronized so that the different intersection exit routes do not interfere with each other [20]. Furthermore, some city authorities require to favor specific urban arterial roads or urban motorways at the cost of worsening the traffic situation on the adjacent smaller roads regardless of their traffic load [53]. Current work regarding the DRL action design does not take into consideration those requirements.

*4.3. Reward Function*

The reward function in RL is used to evaluate the impact of an action on its environment [54]. It is represented by the set of scoring values computed for each action interval. During the learning process, RL must maximize its cumulative reward that agents receive in the long run. In early studies regarding the application of DRL for the ATSC algorithm design, the reward function for one intersection was defined as the change in the cumulative vehicle delay between the actions [16]. Furthermore, in some early studies, the reward was computed as the cumulative waiting time of vehicles measured from the point of time when those vehicles have entered particular intersection bound until the start of green light [21]. Kim and Jeong [48] extended the previously mentioned reward design by using delay instead of waiting time, and simultaneously introduced the concept of *team reward* for several intersections. The mentioned approach is described as follows:

$$r_t = \mathbb{R}(S_t, A_t) = W_t - W_{t+1}. \tag{7}$$

$$W_t = \sum_{i=1}^{N} w_t^i, W_{t+1} = \sum_{i=1}^{N} w_{t+1}^i, \tag{8}$$

$$w_t^i = \sum_{j_t=1}^{M_t} w_{j_t,t}^i, w_{t+1}^i = \sum_{j_t=1}^{M_t} w_{j_{t+1},t+1}^i, \tag{9}$$

where $j_t$ is the the *j*th observed vehicle in the *t*th time step, $M_t$ is the total number of vehicles until the end of *t*th time step, $w_{j_t,t}^i$ is the computed delay of the vehicle *j* at the *i*th intersection until the end of *t*th time step, *N* is the number of intersection at the one region (under condition that traffic network is divided into the several regions), $w_t^i$ is the the sum of delay of all vehicles at the *i*th intersection until the *t*th time step, and $W_t$ is the sum of cumulative delay of all vehicles at all intersections until *t*th time step.

The latest reward functions used for the complex traffic networks are commonly based on the following Key Performance Indicators (KPIs); (1) the average delay, which is computed for all vehicles according to the average waiting time of each vehicle at a single of multiple intersections; (2) the throughput, which is computed as the total traffic flow at one or multi-intersection scenarios; (3) the Average Travelling Time (ATT) of vehicles, which denotes the average traveling time through the one or more intersections; (4) the number of arrival vehicles, which indicates the number of vehicles that

have arrived at their destination through the controlled area in a given period of time [18]; and (5) the number of queuing vehicle at each intersection bound [54].

Liu et al. [55] addressed the problem related to global vehicle delay minimization since the DRL based agent can favor specific TSPs affected by the large traffic load. That might cause an unusual long extension of red signal time for TSP with lower traffic load. To alleviate that problem, the reward function, inspired by the *U.S. Bureau of Public Roads (BPR)* function used for transportation planning, is introduced as follows:

$$r_t = \frac{1}{N} \sum_{i=1}^{N} T_u \left[ \eta - \eta \left( \frac{w^i}{C} \right)^\tau \right]. \tag{10}$$

where $w^i$ is current waiting time for vehicle $i$, $T_u$ is one unit of time, $C$ is the generally acceptable waiting time, and $\eta$ and $\tau$ are constants which are set as $\eta = 0,15$ and $\tau = 2$, respectively, according to the BPR. multi-objective reward function.

Gu et al. [52] concluded that it is hard to compute average delay in real-life traffic environment without the presence of Connected and Autonomous Vehicles (CAVs). It is very difficult to measure sufficiently accurately waiting time for each vehicle at the intersection. The same study proposes a novel and simple approach for reward computation based on the difference between the number of vehicles that passed through the stop line during the last time interval and those that could not pass the intersection and therefore they stayed in the queue. Additionally, the agent for the final reward uses the values of the mentioned differences for each intersection bound but computed for two consecutive time steps. The same idea is indirectly addressed in [22], where Equation (7) is reformulated to use the aggregated queue lengths in two consecutive time steps at each intersection bound. Those approaches require one pair of traffic detectors at each intersection bound for each lane. One of them is installed at the stop line and the other one is placed at the predefined distance (usually 50 m away from the stop line) with the main role to count incoming vehicles to the particular intersection bound. This traffic detector setup is common for signalized intersections so there is no need for additional traffic data sources. In cooperative control, agents primarily compute their local rewards and store them into the reward vector. The shared reward is computed based on this vector. The mentioned reward is used for adjustment of the agent's local rewards. Reward also can be conceptualized as the

In [46], the mentioned reward approach is proposed to maximize the vehicle throughput and minimize the trip delay through the learning process. Those two objectives are represented by the number of vehicles that have entered the intersection, and the waiting time of vehicles at two measuring stations. One station is set near the stop line and the second is placed at a predefined approaching distance, usually 50 m away from the stop line. Both measuring stations are on the same intersection bound. The results of waiting time at the same detection locations in the same TSP are summed and weighted. Finally, those two weighted waiting time values for each bound are subtracted from the weighted value of cumulative vehicles number that has entered those bounds.

The concept of disaggregated reward presented in [20] is based on the idea that the local scalar rewards are computed for each detector in the traffic network. All those rewards are stored into the one vector of a dimension $1xN$, where $N$ is the number of detectors; thus, it is now possible to have $r : SXA \rightarrow \mathbb{R}^N$. In this case, valuable information regarding the relation between the location and effect of taken action is preserved and leveraged by the structure of DDPG. This is analogous to the configuration with $N$ agents that are sharing the same Actor and Critic DNN model weights $\theta^\pi$ and $\theta^Q$, and being trained simultaneously over $N$ different uni-dimensional reward functions. The disaggregated reward can be understood as the novel approach for enabling multi-objective RL.

**Table 2.** The detailed configuration of the most representative DRL frameworks used in the ATSC design.

| DRL Method/Year | DNN Configuration | DNN Parameters [1] | Optimization Algorithm | ER/ Batch Size | Learning Episodes |
|---|---|---|---|---|---|
| Regional A3C + PER 2019 [17]; | Single A3C CNN (1: 3×Conv, 2: 1×FC, 3: 2×LSTM [2]) | 1: (32, 4×4, 2) (64, 2×2, 1) (128, 2×2, 1) 2: 128 3: (128, 128) | AdaBound | 50.000 TS /32 | 1 Million TS; |
| Stabilised IA2C 2019 [9]; | Single A2C 3 inputs per 1×FC, 1×LSTM [3] | 1 input 128 2 input 32 3 inputs 64 LSTM: 64; | Orthogonal initializer, RMSprop as gradient optimiser | 1000 TS /20 ; | 1400 episodes (1 episode is 720 TS); |
| Hierarchical regional A2C 2019 [34]; | Regional A2C agents: 2×(3×MLP+ReLU) [4] Global layer: 1×FC+ReLU | Agent: Critic net: (300, 200, 200) Actor net: (200, 200, 100) | Adam | - /64 | 250 episodes (1 episode is 1000 TS); |
| 3DQN + PER 2019 [46]; | Dueling CNN: 3×Conv 2×FC | 1: (32, 3×15, 3,1) 2: (64, 2×2, 2) 3: (128, 2×2, 1) FCs (464, 64) | Adam | 100.000 TS /32 ; | 1000 episodes [5] |
| ResNet based A2C 2018 [18]; | Single A2C DNN, 4×ResNet Blocks (2×Conv, 2×BN, ReLU) Actor and Critic each: (1×Conv, 1×BN, 2×FC) | Filters for Conv in ResNet (32, 64, 128, 256) | Adam | - /64x× (16 agents) | 50 episodes (1 episode is 3600 TS) |
| DPG 2017 [20] | (4×FC+LeakyReLU, 1×BN+ReLU Gaussian Noise) (4×FC+Leaky ReLU) [6] | Critic: (4×nd + np, 1×nd) Actor: (2×nd + np, 1×np, 1×nd) [7] | Adam | - /- | 1000 episodes |
| DQN + ER 2017 [21]; | Each input 2×Conv merged with 2×FC; | 1: (16, 4×4, 2) 2: (32, 2×2, 2) 3: 128, 4: 64 | RMSprop | 200 episodes /32 | 2000 episodes [5] |
| DQN + ER 2017 [16]; | 2×Conv + 2×FC; | 1: (16, 4×4, 2) 2: (32, 2×2, 2) 3: 128, 4: 64 | RMSprop | 111 simulations /16 | 100 simulations (1 simulation is 4500 TS) |
| DeepSAE + RL 2016 [14] | 4 layer SAE | (32, 16, 4, 2) [1] | SGD | - /- | - |

[1] Convolution layer (Conv) has the notation "(number of filters, filter size, stride)", Fully Connected (FC) layer notation denotes the number of neurons for each layer in an FC stack, LSTM layer notation denotes only an output size, MLP layer stack is denoted by a number of neurons in each layer, and SAE layer notation includes a number of neurons per each layers starting from the input layer. [2] Differentiation of the Actor and Critic models is done by the two separate LSTM layers; the Actor part is designed in the form of Dueling network. [3] Differentiation of Actor and Critic is done by the single LSTM layer; for each of the three inputs, there is one FC layer. [4] These are per each Actor and Critic; their design is based on *Wolpertinger* architecture with DDPG. [5] One episode lasts for 1.5 h. [6] Each set of parameters is for a separate Actor–Critic DNNs. [7] *np* is the total number of phases and *nd* is the number of detectors at the traffic network.

## 5. Open Traffic Data Framework in the Context of Deep Learning

Most recently, a significant increase in the availability of real-time traffic data can be noticed in the complex and ever-expanding traffic environments. This increase corresponds with the two modern trends that are happening in parallel. The first trend is related with a high percentage of smartphone ownership, implementation of innovative traffic sensors, and development of infrastructure for fast and secure data exchange, while the second trend is related with the increase in traffic demand for daily migrations due to a growing number of personal vehicles and traffic network capacity expansion in overpopulated urban regions. Datasets obtained from those evolving traffic environments are nowadays mainly stored at the cloud-oriented *data lakes*. Those datasets contain various traffic data such as GPS trails of vehicles/mobile phone users; vehicle positioning and queue counting at intersections by using advanced video processing [56] and other sensor technologies such as inductive loops, weather, and condition of road surface data; satellite and drone imagery; social network data; etc. The existing traffic datasets will be significantly expanded by the upcoming CAVs.

Those vehicles are equipped with advanced on-board sensors that can generate more accurate traffic data and they have the ability to extend its sensory range by exchanging data with other CAVs and road-side-sensory-units [57].

According to the above, current traffic datasets can be categorized as *big data* considering their volume, variety, and velocity. Traffic *big data* are created based on mobile users data, live traffic video signal processing, CAVs, and other traffic sensors telemetry, and it can complete missing pieces of data through the data fusion process. The DNN models in DRL frameworks dedicated for traffic control require a lot of raw traffic data which must be systematically extracted from different sources, fused, and arranged in the large sets of image-like data structures. This study is oriented towards finding a suitable standardized approach for the image-like state formatting related with traffic flows dynamics at the large traffic networks.

Both image-like representations of intersection network in Figure 5 are recognized as the possible approaches for standardized formatting of raw or aggregated traffic data into a high level image-like network-wide traffic state representation. Those holistic traffic state representations dedicated for deep learning could be directly used by the DNN models or they can be additionally processed. In the context of Open Traffic Data, those standardized image-like data structures can be a potential platform for data sharing among regional data centers interested in network-wide deep learning. Those image-like data structures shared across mentioned data centers will enable a more extensive learning dataset which will enable a more comprehensive, and robust learning. Consequently, this could lead to flexible scalability among various multi-agent DRL approaches applied in ATSC. Additionally, the seamless exchange of image-like state representation at the level of intersection network could enable easier integration between various deep learning applications within the domain of traffic engineering. The whole mentioned data platform is extended up to the novel concept of Deep Open Traffic Data (DOTD) framework, which is illustrated in Figure 6.

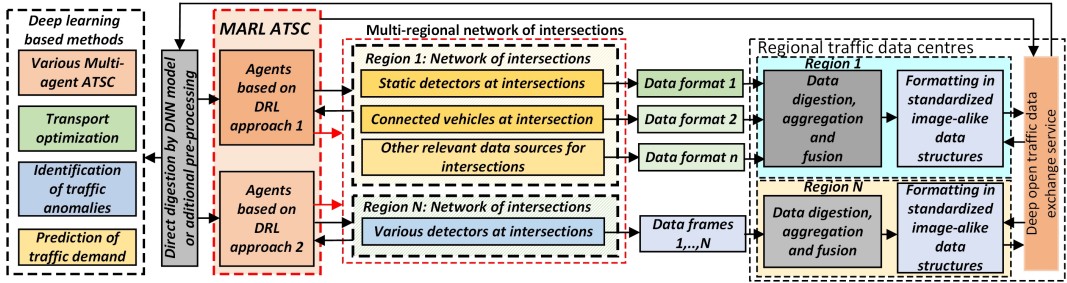

**Figure 6.** The concept of Deep Open Traffic Data framework applied for intersection network.

## 6. Discussion

Fixed traffic signal plans can produce satisfying performance when traffic conditions are consistent in a temporal and spatial context. However, their performance degrades in cases when the traffic conditions are subject to the rapid fluctuations in mobility demand or in the case when intersection throughput is reduced. Those disruptions in traffic flows can occur due to the various types of traffic incidents, public events of great interest, or unanticipated road obstructions. The RL as one of the approaches for on-line machine learning provides an optimal ratio between the complexity and efficiency among various model-free data-driven traffic control methods. It introduces needed self-adaptation features to tackle control problems related to the new unforeseen traffic scenarios.

Conventional RL based approaches store records that contain the described relationship between the states and actions by using a 2-dimensional matrix. In the case of a large traffic network, this matrix or table can become too vast to learn. An estimation of the return with respect to the mentioned table is introduced and it is known as the value function [28]. Those types of RL algorithms are known as Critic-only. They depend on the efficiency of value function approximation since they do not use the explicit function for the policy. The RL algorithms based on the policy function are known as

the Actor-only algorithms and they use a large continuous action spectrum. The drawback of those algorithms is high variance in the gradient estimation what makes their learning slow [28]. There are numerous methodologies that can reduce the high-dimensionality of Critic-only RL algorithms by conducting the value function approximation schemes in a form of an empirical balance between representational power and computational costs such as Tile Coding and Radial Basis Functions (RBF). However, they still require manual feature extraction from the raw traffic data. During this process, it is possible to lose a lot of useful information about traffic flow. The introduction of DNN models that utilize the feature extraction and estimation/classification has enabled direct learning from the information-dense input data. Furthermore, the use of the DNN model in the DRL framework as the function approximator within a Critics-only RL setting removed the need for searching vast tables what consequently mitigate the problem with dimensionality explosion. Currently, the most used architecture for the DRL framework is based on the Actor–Critic approach since the Critic part evaluates the quality of the used policy and the Actor uses the Critic's information to update its policy parameters. Those two integrated parts are capable of producing continuous actions while the high variance in the estimation of gradients of Actor-only methods is reduced by considering the value function of the Critic part as it is represented in [58].

The main efficiency measure for novel ATSC algorithms is their effectiveness in holistic traffic signal control since the poor scalability is the most prominent drawback of those types of algorithms [59]. During the last five years, according to Table 1, it is possible to notice an increase in the number of intersections that ATSC based on the DRL approach can handle. The scalability of the latest ATSC approaches which are based on DRL is tackled by the deep MARL frameworks. In most cases, those frameworks are based on the global Q-function distribution to the local DRL agents in the form of the IA2C, transfer learning, or introducing a hierarchical regional A2C/A3C approach. Simultaneously, over the past five years, it is possible to notice the growing complexity of DNN models which are used in the advanced Actor–Critic architectures. All mentioned advancements are proof of the growing interest of the scientific community in that field. Furthermore, most recent research results achieved in the ATSC based on the multi-agent DRL are compared not just with the traditionally used ATSC algorithms but with the other inferior DRL approaches. The growing maturity of DRL applications in ATSC is additionally proven by the transition from the synthetic traffic simulations towards the simulations which are tuned by using real-world traffic data (e.g., the Monaco traffic network use case study presented in [9] and the Sants area case study in the city of Barcelona presented in [20]).

The most noticeable limits of DRL frameworks applied in ATSC are the following: (1) slow and computationally expensive learning; (2) difficulties in optimal reward design for stochastic control environments, e.g. multi-intersection network; (3) over-fitting to unusual patterns which can be common in traffic flow, e.g. effects of aggressive overtaking or lane changing, impacts of lesser incidents, etc.; (4) absence of explanation how the control decision was made; and (5) control decision can be unstable and hard to reproduce. Additionally, the DNN models in DRL frameworks can digest only adequately pre-processed raw traffic data. The efficiency of DRL framework strongly depends upon the positional arrangement of raw data in image-like input structures suitable for the DNN model digestion. A large network of intersections require aggregation of raw traffic data up to a certain level in order to be arranged in the image-like format which can represent overall traffic state at such networks. Furthermore, all obtained traffic data should be processed/formatted by using the standardized image-like format and be freely shared in line with the Open Traffic Data concept. The process of sharing should be done through the local traffic data centers to achieve flexible and broader scalability over various DRL oriented ATSC approaches. The DOTD as a conceptual framework for processing and exchanging standardized higher-level image-like formats will be essential for more extensive application of DNN based approaches in the traffic engineering such as prediction of traffic demand [60,61], detection of traffic anomalies [62], optimization of transportation processes [63], and application of DRL in real-world traffic control problems [64]. The DOTD framework potentially

can be a data platform for establishing interoperability between all mentioned traffic engineering fields related to deep learning.

## 7. Conclusions

The core part of this paper is an overview of the recent applications of DRL in ATSC. Although the application of DRL in ATSC is a relatively new field, the papers reviewed show promising results compared to the traditional ATSC algorithms and ATSC algorithms that are based on RL methodologies. During the last five years, it is possible to notice an increase in the number of intersections controlled by the use of various multi-agent frameworks based on DRL. Transfer learning, multi-agent frameworks, and complex DNN model structures based on the Actor–Critic architecture are the most prominent approaches for improving the effectiveness of the DRL in the holistic traffic signal control. The development of the advanced DRL frameworks for large-scale ATSC is proof that the scientific community has recognized this approach as a promising direction for future traffic signal control. Furthermore, this study presents recent advancements in the state, action, and reward modeling, which is currently a very open area for research. Additionally, this study provides an assessment of the issues concerning the relationship between the learning convergence, DNN model complexity, data availability, and scalability for large intersection networks. Feasible solutions to those issues are key for a possible real-world application of those complex traffic signal control frameworks. Standardized image-like state representation is found to be important for interoperability and flexible scalability between the various ATSC approaches that are based on DRL. The availability of the larger traffic data spectrum generated by the various sensors and upcoming CAVs will create information-dense traffic data environments. Data from those environments must be obtained in a centralized fashion, pre-processed, and formatted up to a higher level of image-like traffic state representation. Each application of DRL in ATSC can directly use those image-like data structures or additionally process them in order to be digested by the used DNN model. This paper provides several existing DTSE approaches suitable for initial high-level pre-processing of the raw traffic data. The legislative and technical framework for gathering and sharing mentioned pre-processed traffic data must be in line with the Open Traffic Data concept in order to enable network-wide, inter-operable, and multi-agent ATSC based on information dense state representation. Future improvements in DRL frameworks for ATSC are expected to be based on the integration of Deep MARL approaches with multi-objective reward function and transfer learning to address the problem regarding the multi-intersection signal control. Mentioned complex DRL frameworks are expected to be optimized by the multi-objective evolutionary algorithms that can potentially enable integrated control between multi-intersection traffic signal control and traffic control methods at nearby motorway (e.g., ramp metering, variable speed limit control, etc.). The DOTD as the conceptual approach is expected to provide network-wide tensor-based data structures that will integrate fused traffic data, weather data, and road condition data in order to proved robust and more comprehensive inputs for advanced DRL frameworks applied in ATSC.

**Funding:** This research has received funding from the European Union's Horizon 2020 research and innovation program under grant agreement No. 857592.

**Acknowledgments:** This research was part of a project Twinning Open Data Operational that has received funding from the European Union's Horizon 2020 research and innovation programme under grant agreement No. 857592.

**Conflicts of Interest:** The authors declare no conflict of interest.

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
