# Peer review of "Application of Deep Reinforcement Learning in Traffic Signal Control: An Overview and Impact of Open Traffic Data"

_applsci, doi:10.3390/app10114011_

Round 1

Reviewer 1 Report

This paper is an overview of the recent applications of Deep Reinforcement Learning in Traffic Signal Control.
It is clear from the paper that the introduction of these techniques is fundamental and necessary to deal, with the advanced traffic control methods, the persistent problems with intense congestions and their negative impact on sustainablemobility.
To think of providing an exhaustive treatment to such an evolving context is unthinkable but the authors have clearly provided state of the art.
The work perhaps lacks in some cases more critical emphasis that would have helped to understand the limits of the state of the art.
There is also a lack of clear hints of future improvements that could be expected.

Reviewer 2 Report

The authors present a study focused on the control of the traffic signal since the impact of congestion is most notable in the dense urban traffic networks using the machine learning methodology for the control of traffic signals called Reinforcement Learning. This article is scientifically well written and consistent. Therefore, I will detail some suggestions for improving this document: -In the presentation of the Sections, Section 5 has not been included. -What would happen if the architecture for traffic signal control is decentralized or centralized when applying reinforcement learning? -The authors should argument about the different levels of traffic simulation (microscopic, mesoscopic and macroscopic) with a brief introduction. -For more arguments about SUMO regarding the implementation of real traffic that is discussed in Section 4. I recommend this article: ==> Zambrano-Martinez, J. L., Calafate, C. T., Soler, D., & Cano, J. C. (2017). Towards realistic urban traffic experiments using DFROUTER: Heuristic, validation and extensions. Sensors, 17 (12), 2921. -What innovative sensors can be implemented in traffic to capture traffic behavior? Some sensors should be named as the Induction Loops. -I suggest an article that relating obtaining traffic data through video processing as mentioned in Section 5: ==> Kastrinaki, V .; Zervakis, M .; Kalaitzakis, K. A survey of video processing techniques for traffic applications. Image Vis. Comput. 2003, 21, 359–381, doi: 10.1016 / S0262-8856 (03) 00004-0. -Authors should write about future works.
